# Intermedilysin cytolytic activity depends on heparan sulfates and membrane composition

Gediminas Drabavicius[1,2], Dirk Daelemans[1]*

**1** KU Leuven Department of Microbiology, Immunology, and Transplantation, Laboratory of Virology and Chemotherapy, Rega Institute, Leuven, Belgium, **2** Vilnius University, Life Sciences Center, Institute of Biotechnology, Vilnius, Lithuania

* dirk.daelemans@kuleuven.be

**Data Availability Statement:** All relevant data are within the manuscript and its Supporting Information files.

**Funding:** The author(s) received no specific funding for this work.

## Abstract

Cholesterol-dependent cytolysins (CDCs), of which intermedilysin (ILY) is an archetypal member, are a group of pore-forming toxins secreted by a large variety of pathogenic bacteria. These toxins, secreted as soluble monomers, oligomerize upon interaction with cholesterol in the target membrane and transect it as pores of diameters of up to 100 to 300 Å. These pores disrupt cell membranes and result in cell lysis. The immune receptor CD59 is a well-established cellular factor required for intermedilysin pore formation. In this study, we applied genome-wide CRISPR-Cas9 knock-out screening to reveal additional cellular co-factors essential for ILY-mediated cell lysis. We discovered a plethora of genes previously not associated with ILY, many of which are important for membrane constitution. We show that heparan sulfates facilitate ILY activity, which can be inhibited by heparin. Furthermore, we identified hits in both protein and lipid glycosylation pathways and show a role for glucosylceramide, demonstrating that membrane organization is important for ILY activity. We also cross-validated identified genes with vaginolysin and pneumolysin and found that pneumolysin's cytolytic activity strongly depends on the asymmetric distribution of membrane phospholipids. This study shows that membrane-targeting toxins combined with genetic screening can identify genes involved in biological membrane composition and metabolism.

## Author summary

Bacterial toxins are one of the principal tools pathogenic bacteria use to compromise hosts. Interactions of pathogenic toxins with human cells have been extensively studied. The advent of genetic approaches such as genome-wide CRISPR knock-out screening has allowed researchers to unravel novel mechanisms in different fields of research. This has led to unprecedented growth in knowledge about a wide variety of biological phenomena. Intermedilysin, a bacterial pore-forming toxin, is known to require cholesterol and the cellular receptor CD59 for its cytolytic activity, and there is little knowledge of other cellular vulnerabilities. Here, we applied genome-wide functional genetic screening to identify genetic dependencies of intermedilysin. We identified many genes and pathways that have not been previously associated with intermedilysin, such as heparan sulfate and

**Competing interests:** The authors have declared that no competing interests exist.

genes involved in membrane organization. The findings described in this study are particularly relevant to the advancement of our understanding of bacterial pore-forming toxins, in addition to revealing genetic details of membrane domain formation, which are targeted by toxins.

## Introduction

Bacteria use a variety of strategies to exploit hosts, and one of the prominent ways to achieve this is by toxins. Pore-forming toxins are a group of bacterial membrane-targeting toxins that pathogens use to invade niches[1], release nutrients[2], or subvert the host's defenses[3]. Unique among pore-forming toxins are cholesterol-dependent cytolysins (CDCs), which are the main virulence factors of a variety of pathogenic bacteria[4–8] that form pores (~100–300 Å diameter). CDCs belong to the MACPF/CDC (membrane attack complex-perforin/cholesterol-dependent cytolysin) protein family used by both the immune system and pathogens to lyse target cells by forming large membrane pores[9, 10]. Cells can repair membrane disruption caused by toxins through exocytosis or shedding of the damaged membranes[11], and some cells are more resistant to these toxins than others[12, 13]. Because of their inherent ability to bind to the membranes, non-toxic mutants of CDCs are used to study membrane domains[14–16].

Typically, CDCs only need to bind to membrane-embedded cholesterol to form pores and transect membranes. However, a small group of CDCs depend on human glycosylphosphatidylinositol (GPI)-anchored protein CD59[4, 17]. Nonetheless, only one known CDC, namely, intermedilysin (ILY), cannot function without CD59. ILY is secreted by pathogenic strains of *Streptococcus intermedius* and is a potent cytotoxin[18]. Additionally, its strict requirement for human CD59 and inability to lyse murine cells, for example, explains its unique tropism for human cells[18, 19]. Consequently, this feature of the toxin has been used in cell ablation studies to understand cell lineage and function. Expressing human CD59 under tissue-specific promoters and injecting mice with ILY allows ablating entire cell populations rapidly[20]. Interestingly, the typical function of CD59 is to prevent membrane attack complex assembly on cells. This mechanism enables the complement arm of the immune system to differentiate between self and foreign cells, as foreign cells do not express CD59[21]. Thus, ironically, to form pores, ILY uses the same protein factor which prevents the pore formation of the membrane attack complex.

Several recent studies demonstrated the activity of CDCs may depend on additional factors such as various glycans[22, 23]. Moreover, some have speculated that most, if not all, CDCs possess a lectin-like activity[22, 23]. Previous studies have used glycan libraries to screen CDC-glycan interactions. Recently, genome-wide CRISPR knock-out screening has been successfully used to identify cellular factors and processes involved in cell susceptibility to bacterial toxins[24–27]. In this study, we used this screening approach to identify factors required by ILY. Additionally, we hypothesized that CRISPR screening using membrane binding toxins would teach us about membrane component metabolism and, ultimately, membrane composition. We identified many genes not previously associated with ILY in addition to confirming the role of CD59 and cholesterol. Furthermore, we established the importance of glycosylation and heparan sulfates, specifically for ILY. Moreover, some genes identified in our ILY screen proved to be important in the mechanism of action of other CDCs as well. Lastly, we found genes involved in membrane component synthesis and metabolism, showing that differential membrane composition is important to ILY and other CDCs. Therefore, the combination of

CDCs or other membrane binding toxins and genome-wide CRISPR screens can be used to study the cell-toxin interactions and the synthesis, metabolism, and trafficking of membrane components.

## Results and discussion

### Intermedilysin genome-wide CRISPR knock-out screen

To identify genes essential for ILY-mediated toxicity, we applied a genome-wide CRISPR knock-out screening approach. We transduced the genome-wide CRISPR Toronto knock-out library (TKOv3)[28] in near-haploid HAP-1 cells at a multiplicity of infection (moi) of ~0.3 to generate a pool of cells bearing single gene knock-out. Transduced cells were enriched by puromycin selection, and a sample was collected to measure the guide representation in the cell population before ILY selection. The remainder of the cells was treated with 2 ng/ml ILY. ILY treatment induced severe cytotoxicity within one hour of treatment. After allowing the surviving cells to expand for two weeks, a sample of cells selected with 2 ng/ml ILY was collected to quantify guide representation in this population. Then, these cells were subjected to the second round of ILY selection with 10 ng/ml ILY, and cells were allowed to expand again, after which we collected another sample. Next, genomic DNA from cells before ILY selection and after ILY selection was extracted and sequenced for sgRNA abundance. Guide representation in the treated conditions was calculated as $\log_2$-fold change compared to the guide representation in the condition before treatment (Fig 1A). We ran a similar screen in parallel using the well-characterized diphtheria toxin (DT) and did not observe any commonalities between the screens, eliminating innate library bias as a potential source of hits (S1 and S2 Figs, S1 Table). For DT, the identified hits were in agreement with the literature[26, 29]. Specifically, knock-out of *HBEGF*, which encodes a DT receptor, or knock-out of genes in the diphthamide synthesis pathway conferred resistance to DT (S1 Fig and S1 Table).

Hits from the 2 ng/ml ILY screen included CD59 and genes associated with GPI-anchor synthesis and attachment, as well as a myriad of newly identified genes that were previously not associated with ILY (Figs 1B and 1C and S3, S1 Table). Genes were designated as a hit if they were enriched at least four times in one of the two replicate screens and had a p-value of >0.01. *CD59* and GPI-anchor synthesis genes (20 hits) were classified as strong hits since these remained after increasing ILY concentration to 10 ng/ml (S2 Fig). The weaker hits are the ones that were present in the 2 ng/ml ILY treatment but disappeared when cells were treated with 10 ng/ml. These include genes from nucleotide sugar synthesis (3 hits), lipid and protein glycosylation (10 hits), heparan sulfate (6 hits), cholesterol metabolism pathways (6 hits) as well as some genes with unknown function (14 hits) (Figs 1B and 1C and S3).

To validate the hits identified in our screen, we separately knocked them out in wild-type HAP-1 cells using guides different than those used in the screen, namely the respective guide RNAs from the Brunello genome-wide knock-out library[30] (S2 Table). For the strong hits, we chose to validate *CD59* and only one gene from the GPI-anchor synthesis cascade, *PIGA*. Knock-out of *CD59* and *PIGA* (the first gene in GPI-anchor synthesis cascade) both conferred complete resistance to ILY (Fig 1D) due to the depletion of CD59.

For the weaker hits, all genes were separately knocked out, and resistance to ILY treatment was measured (Fig 2A). From the 37 hits identified in the screens, 28 could be validated. From these, two genes involved in cholesterol metabolism and sequestration were identified, namely, *SREBF2*, the master regulator of cholesterol metabolism, and *LDLR* (Figs 1B, 2A and S3). This is in agreement with previous knowledge as ILY, along with other CDC family members, cannot function without cholesterol, and its function is impeded with the reduction of membrane cholesterol[31]. Our screen was able to identify factors that were already known to be

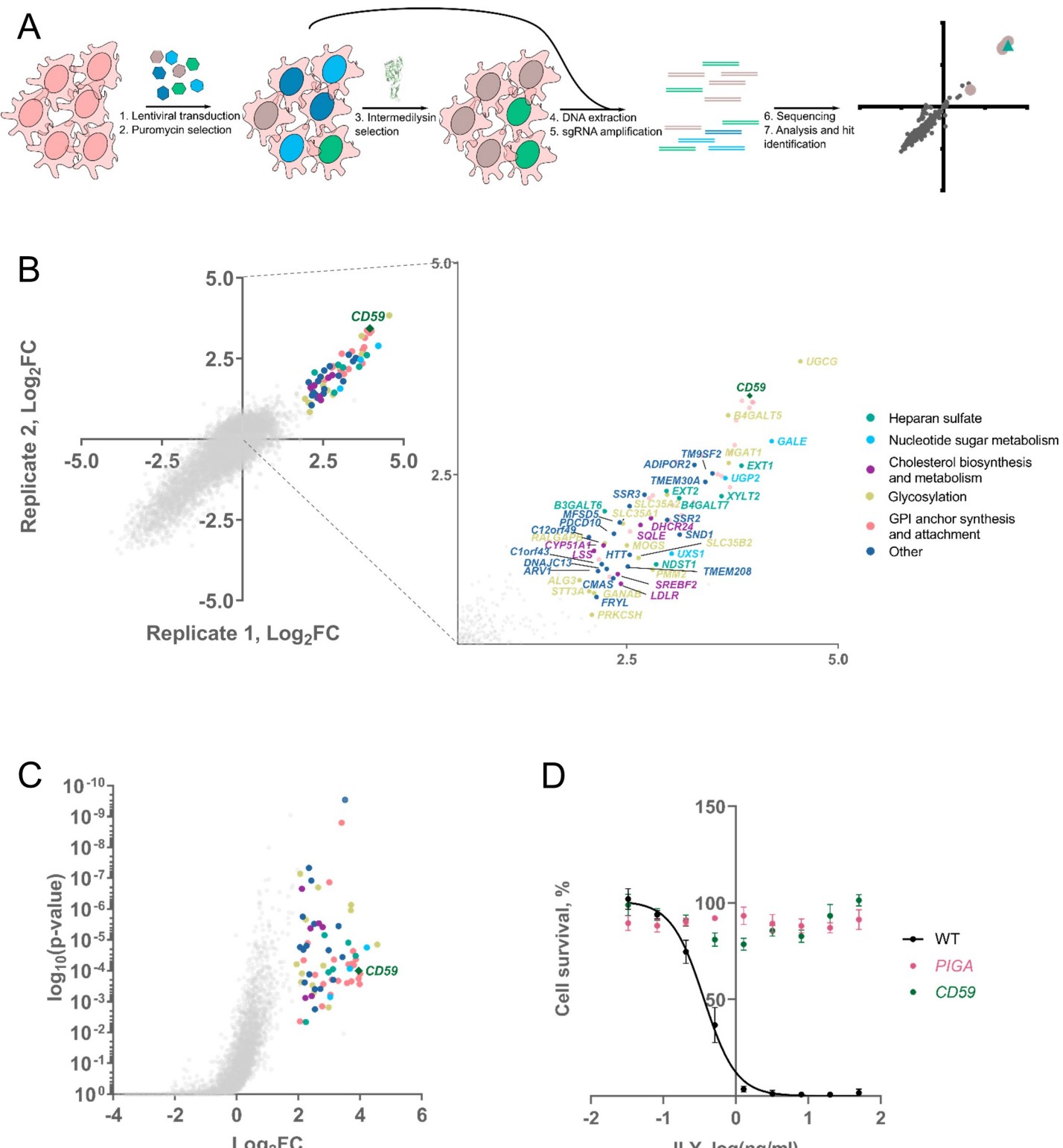

**Fig 1. Intermedilysin genome-wide CRISPR knock-out screening identifies novel genes involved in intermedilysin-mediated cell lysis. A.** Schematic representation of the screen. The population of the cells was transduced with lentiviral vectors harboring the genome-wide knock-out library. After selecting transduced cells with puromycin, we began the selection of transduced cells with ILY. We collected cells before and after selection, extracted DNA, amplified guide sequences nascent in cells, and sequenced them. Afterward, we compared read counts of guides, and after statistical analysis, we identified hits. **B**. Results of intermedilysin CRISPR knock-out screen. Axes represent $Log_2$-fold changes in the average number of guide reads (of all four guides) when compared with non-selected cells from two independent

replications of the screen. We identified GPI anchor synthesis and attachment cascade and *CD59* genes, previously known to be required for ILY-mediated cytotoxicity. Furthermore, we found numerous genes previously not observed to be important in the action of ILY. The panel on the right represents a blow-up of the screen results with all of the identified genes listed, excluding GPI-anchor synthesis, where every member in the pathway was identified. **C**–Volcano plot of a representative 2 ng/ml ILY genome-wide CRISPR screen replicate. The screen was performed in duplicate. Y-axis displays the significance of the hit, while the X-axis represents $Log_2$-fold change in the Average number of guide reads (of all four guides) when compared with non-selected cells. Genes were considered a hit if they were enriched at least four times ($Log_2FC \geq 2$) in one of the replicates and had a p-value of $>0.01$. **D**. *CD59* or *PIGA* knock-out protects cells from the action of ILY. MTS assay was used to determine the percentage of cells surviving the addition of different amounts of ILY. n = 3, error bars represent standard deviations.

important for the action of ILY, such as CD59 and cholesterol. These findings demonstrate that our screen can reveal known cellular factors involved in ILY activity, which can be seen as positive controls for the screen. This strongly supported the further investigation of the other identified hits discovered in our screen.

## Defects in N-glycosylation and ER translocation reduce plasma membrane levels of CD59

First, we investigated whether knock-out of the hit genes identified in the screen reduced plasma membrane levels of CD59, which could explain their mechanism of resistance to ILY. To do that, we stained the validated knock-out cell lines with the APC-conjugated anti-CD59 antibody and assessed CD59 levels by flow cytometry (Fig 2B). While most cell lines contained similar amounts of CD59 in the plasma membrane, N-glycosylation (*MOGS*, *PRKCSH*, *GANAB*, and *MGAT1*) and *SSR1*, *SSR2*, and *SSR3* knock-out cell lines expressed significantly reduced levels of CD59, explaining their resistance to ILY. The reason why the perturbations of the genes involved in N-glycosylation resulted in the reduction of CD59 is not entirely apparent. While CD59 is usually heavily glycosylated, it only contains one single N-glycosylation site. The loss of this particular glycosylation site has been shown to strengthen the complement inhibition[32]. However, there is no data available on its effect on CD59 interaction with ILY. Therefore, it is unclear whether the reduction in plasma membrane CD59 is caused by a specific loss of CD59 N-glycosylation or by the global defect in glycosylation. The reason for the reduction of CD59 in the plasma membrane in the case of *SSR1*, *SSR2*, and *SSR3* perturbation is clearer. SSR1, SSR2, and SSR3 proteins form the translocon-associated complex (TRAP), which plays a role in protein translocation to the endoplasmic reticulum (ER)[33], and hence membrane localization, explaining the loss of membrane CD59 upon knock-out.

## Nucleotide sugars are building blocks for intermedilysin's cellular glycan co-factors

Having identified hits with clear resistance mechanisms, we went on to investigate other groups of hits. The first group of newly identified hits were genes involved in the synthesis of UDP-sugars. Namely, UDP-glucose, UDP-galactose, and UDP-xylose by *UGP2*, *GALE*, and *UXS1*, respectively (Fig 1B). UDP-sugars serve as building blocks for the glycosylation of lipids and proteins. When glucose is taken up, the UDP-glucose pyrophosphorylase enzyme (product of UGP2 gene) activates it by adding a UDP-moiety to the glucose molecule. UDP-glucose can then be converted to UDP-galactose and UDP-xylose by GALE and UXS1, respectively (S4 Fig)[34]. Therefore, the loss of either *GALE* or *UGP2* results in the loss of UDP-galactose. However, in the presence of galactose in the growth medium, cells do not require UGP2 or GALE to produce UDP-galactose (S4 Fig)[34]. The media in which we grow cells only contains glucose as a source of carbon. Thus to independently confirm our hits in UDP-sugar metabolism, we supplemented growth media of *UGP2*, *GALE*, and *UXS1* knock-out cell lines with galactose and measured their susceptibility to ILY. Supplementing growth medium with

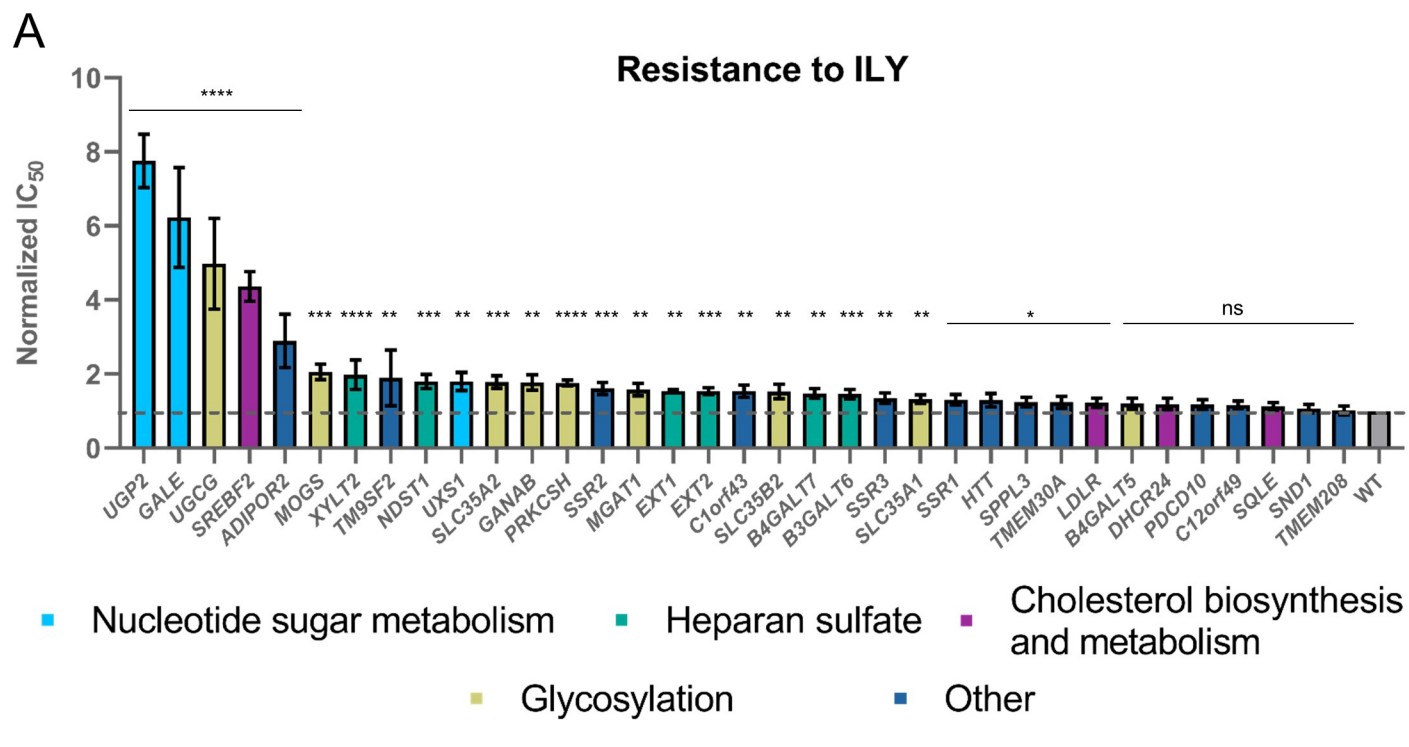

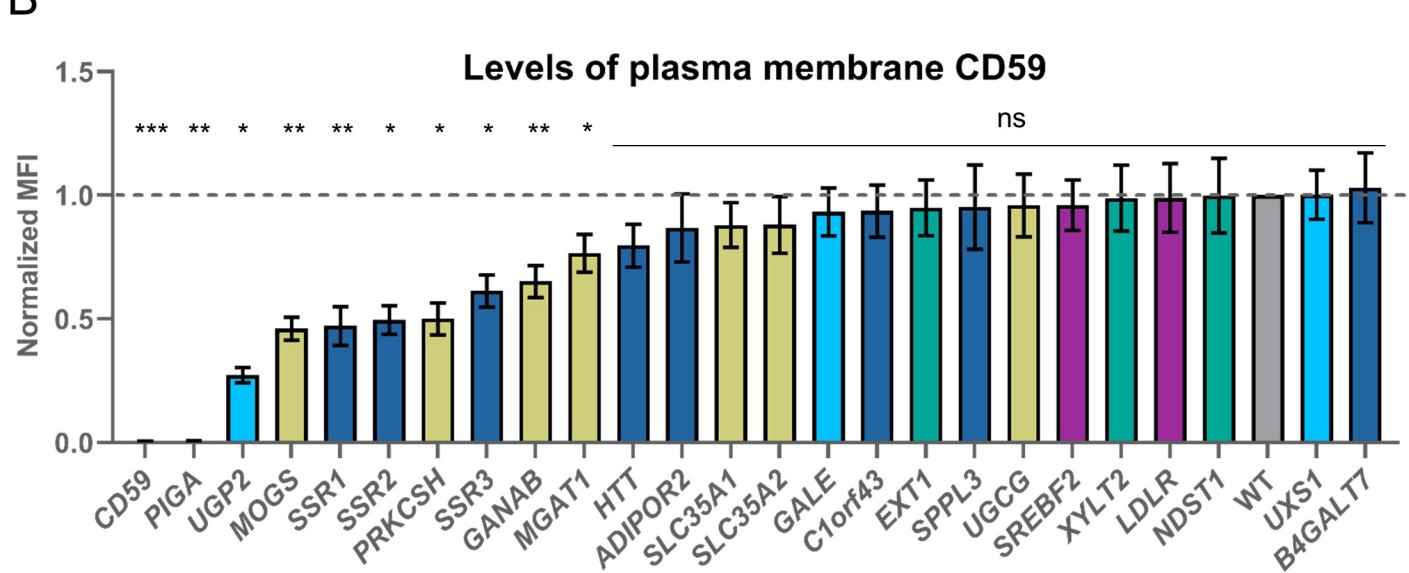

**Fig 2. Validation of the hits from the intermedilysin CRISPR knock-out screening. A.** Resistance to ILY conferred by the knock-out of all tested genes. Values were normalized by dividing the raw $IC_{50}$ value by the $IC_{50}$ value for ILY in WT HAP-1 cells. The dashed line shows the WT $IC_{50}$ value. n = 3, error bars represent standard deviations. p-values were calculated using two-tailed t-test; *–<0.05; **–<0.01, ***–<0.001; ****–<0.0001. Significance was calculated in relation to the WT cell line. **B**. CD59 protein is depleted in *CD59* and *PIGA* knock-out cell lines and significantly reduced in N-glycosylation mutant cell lines as well as TRAP complex gene (*SSR1, SSR2,* and *SSR3*) knock-out cell lines. Cells were labeled with anti-CD59 antibody (OV9A2) conjugated with APC and FACS sorted. Median fluorescence intensity (MFI) in the APC line was compared between WT and knock-out cell lines. Values were normalized by dividing the raw MFI value of a knock-out cell line by the raw MFI value for WT HAP-1 cells. The dashed line shows the WT normalized MFI value. Error bars represent standard deviations from the mean of the medians of at least two technical replicates. Each median was calculated from at least 15000 events. p-values were calculated using two-tailed t-test; *–<0.05; **–< 0.01, ***–<0.001.

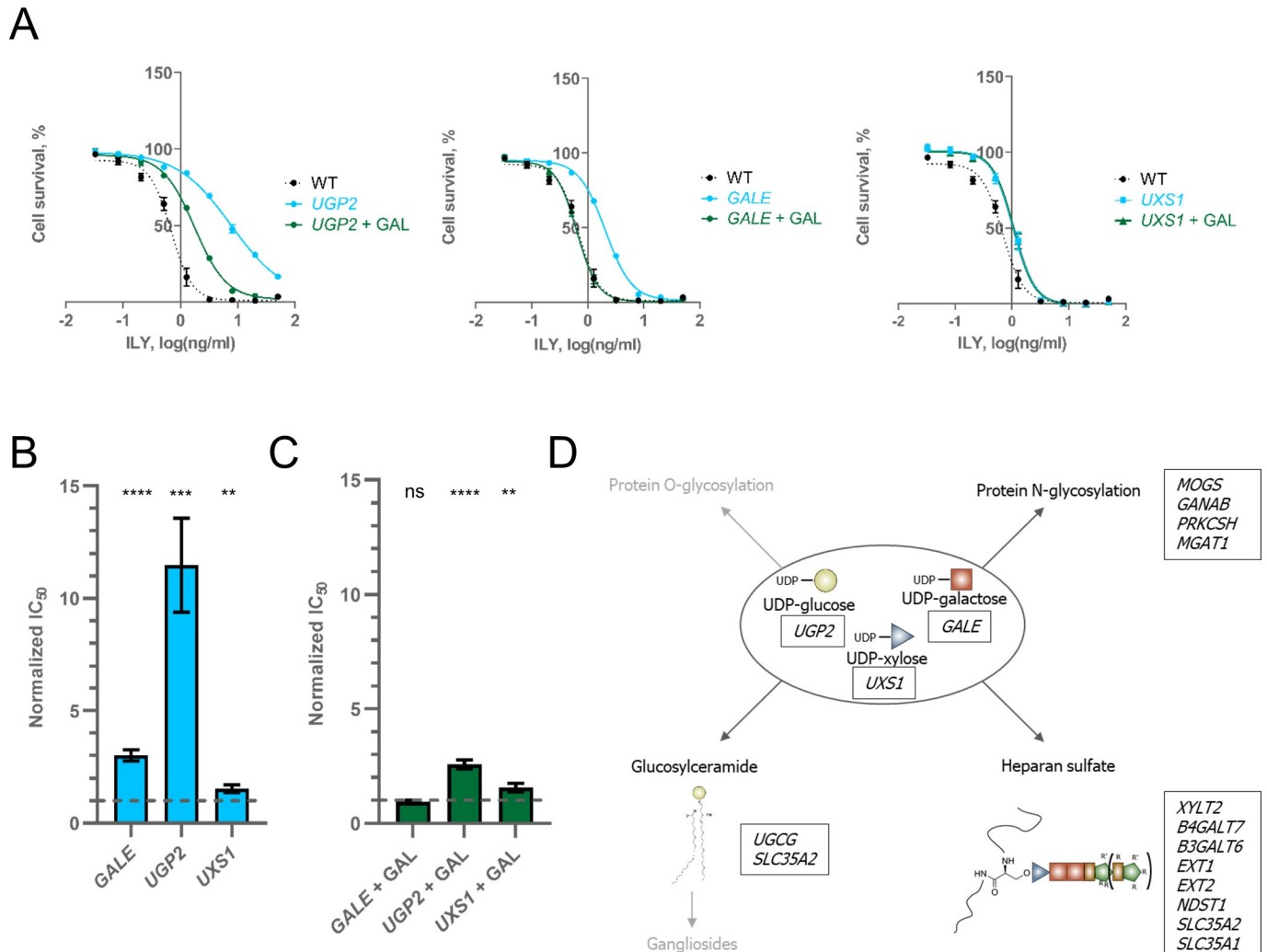

**Fig 3. UDP-sugar synthesis gene knock-outs protect against ILY. A**. Lysis profiles of cell lines with UDP-sugar gene knock-outs before and after growing them in a galactose enriched medium. n = 3, error bars represent standard deviations. **B**. Resistance conferred by the knock-out of *GALE*, *UGP2*, and *UXS1* genes. Values were normalized by dividing the raw $IC_{50}$ value by the $IC_{50}$ value of ILY in WT HAP-1 cells. The dashed line shows the WT $IC_{50}$ value. n = 3, error bars represent standard deviations. p-values were calculated using two-tailed t-test; *−<0.05; **−<0.01, ***−<0.001; ****−<0.0001. **C**. Resistance conferred by the knock-out of *GALE*, *UGP2*, and *UXS1* genes when cell media was supplemented with galactose. Values were normalized by dividing the raw $IC_{50}$ value by the $IC_{50}$ value of ILY in WT HAP-1 cells. The dashed line shows the WT $IC_{50}$ value. N = 3, error bars represent standard deviations. p-values were calculated using two-tailed t-test; *−<0.05; **−<0.01, ***−<0.001; ****−<0.0001. **D**. Schematic representation of glycosylation pathways that use UDP-sugars identified in our screen. Pathway names listed in black represent pathways we studied in depth. Pathway names listed in faded grey represent processes that use UDP-sugars, but no genes from their pathways were identified in our screen. Boxes next to process names enumerate genes in that pathway identified in our screen (proteins produced from listed genes do not necessarily use these UDP-sugars themselves).

galactose completely restored ILY susceptibility of the *GALE* knock-out cell line and restored most of the sensitivity of the *UGP2* knock-out, but did not affect *UXS1* knock-out cells (Fig 3A–3C). Therefore, most of the resistance conferred by the *UGP2* knock-out is caused by the loss of UDP-galactose and not UDP-glucose. The remainder of the resistance provided by the loss of UGP2 can be ascribed to either the loss of N-glycosylation or the reduction of CD59 in the membrane (Fig 2B). These findings highlight the importance of UDP-galactose in cell susceptibility to ILY-mediated lysis and identify *GALE* knock-out as a crucial resistance mechanism. Glycosylation pathways dependent on UDP-galactose and other UDP-sugars are summarized in Fig 3D.

## Heparan sulfate enhances the rate of intermedilysin pore formation

In addition to the genes involved in the synthesis of UDP-sugars, we also identified genes in several glycosylation pathways. For example, UDP-xylose plays a vital role in protein O-glycosylation. Certain types of O-glycosylation, most notably glycosaminoglycans (GAGs), are initiated by the attachment of UDP-xylose to a serine or threonine residue of target proteins by XYLT1 or XYLT2 UDP-xylosyl transferases[35, 36] (Fig 4A). Interestingly, *XYLT2* was identified as a hit in our screen, suggesting that O-glycosylation may be important for ILY action. We also identified additional genes involved in the heparan sulfate (HS) pathway, namely, *NDST1*, *EXT1*, *EXT2*, *B3GALT6*, *B4GALT7*. This finding points towards a role of HS in the action of ILY (Fig 4A). HS is a sulfonylated linear polysaccharide present on some membrane proteins, and it regulates a wide variety of biological functions. Knock-out of *XYLT2* conferred a similar level of resistance as that of *UXS1* knock-out (UDP-xylose synthesis) (Fig 4B and 4C), which suggests that knock-out of *UXS1* protects cells due to loss of HS, especially since the only pathway that depends on UDP-xylose identified in our screen is the heparan sulfate synthesis pathway. Also, part of the resistance conferred by *GALE* and *UGP2* knock-outs may be explained by the loss of HS because UDP-galactose is required by the protein products of B4GALT7 and B3GALT6 to synthesize HS (Fig 4A). Additionally, *B4GALT7* and *B3GALT6* were also identified as hits in our screen.

Although genes involved in HS synthesis were identified and validated, knock-outs conferred only a minor increase in $IC_{50}$ values (Figs 2A and 4B). However, the assay we used to measure resistance or susceptibility to ILY is an end-point assay and does not provide kinetic information. To measure subtler differences, we developed an assay that allowed to observe ILY pore formation kinetics in near real-time. We added propidium iodide to the ILY-mediated lysis reaction and measured the increase in fluorescence intensity while propidium iodide enters the cells through ILY-damaged cell membranes. Pores formed more slowly in cells containing *XYLT2*, *UXS1*, or *EXT1* knock-out than in the WT cell membranes (Fig 4D). To independently confirm the involvement of HS in the mechanism of ILY-mediated cytolysis, we treated cells with heparinases to remove HS before the addition of ILY. Heparinase treatment also impeded pore formation (Fig 4E).

Similarly, adding heparin or free HS, but not other GAGs, to the reaction effectively slowed down pore formation (Figs 4F and S5), demonstrating that heparin and, at higher concentrations, HS can inhibit ILY competitively. These results suggest that ILY binds a structural motif common between HS and heparin to facilitate its activity. While our screen identified each of the genes involved in HS synthesis to confer cells with resistance to ILY, the mechanism for this resistance remains unclear. We hypothesize that HS acts as a first docking site for ILY to increase the probability of binding to CD59. In the first published crystal structure of ILY, sulfate ions were bound in a position that appeared to stabilize the protein[5] (S6 Fig). It was suggested that these sites might represent pockets for a, at that time unknown, protein receptor–human CD59. We hypothesize that these sulfate ions may come from HS. This hypothesis is strengthened by the fact that other GAGs, namely chondroitin sulfate (CS) and hyaluronic acid (HA), did not inhibit the pore formation (Figs 4F and S5). Heparin and HS can exhibit different levels of sulfonylation, with heparin being generally more heavily sulfonylated. CS is, in contrast to heparin or HS, only O-sulfonylated, and HA is a non-sulfonylated. This suggests that the N-sulfonylated glucuronic acid of heparin or HS must be at least partially responsible for the interaction with ILY (Fig 4G).

Additionally, it is tempting to speculate as to the physiological role of heparin. Naturally, heparin occurs at very low concentrations and in select tissues, so its role in the organism is probably different from anticoagulation often ascribed to it[37]. At high concentrations ILY can lyse cells independently of HS (S2 Fig). However, there is little doubt that physiological

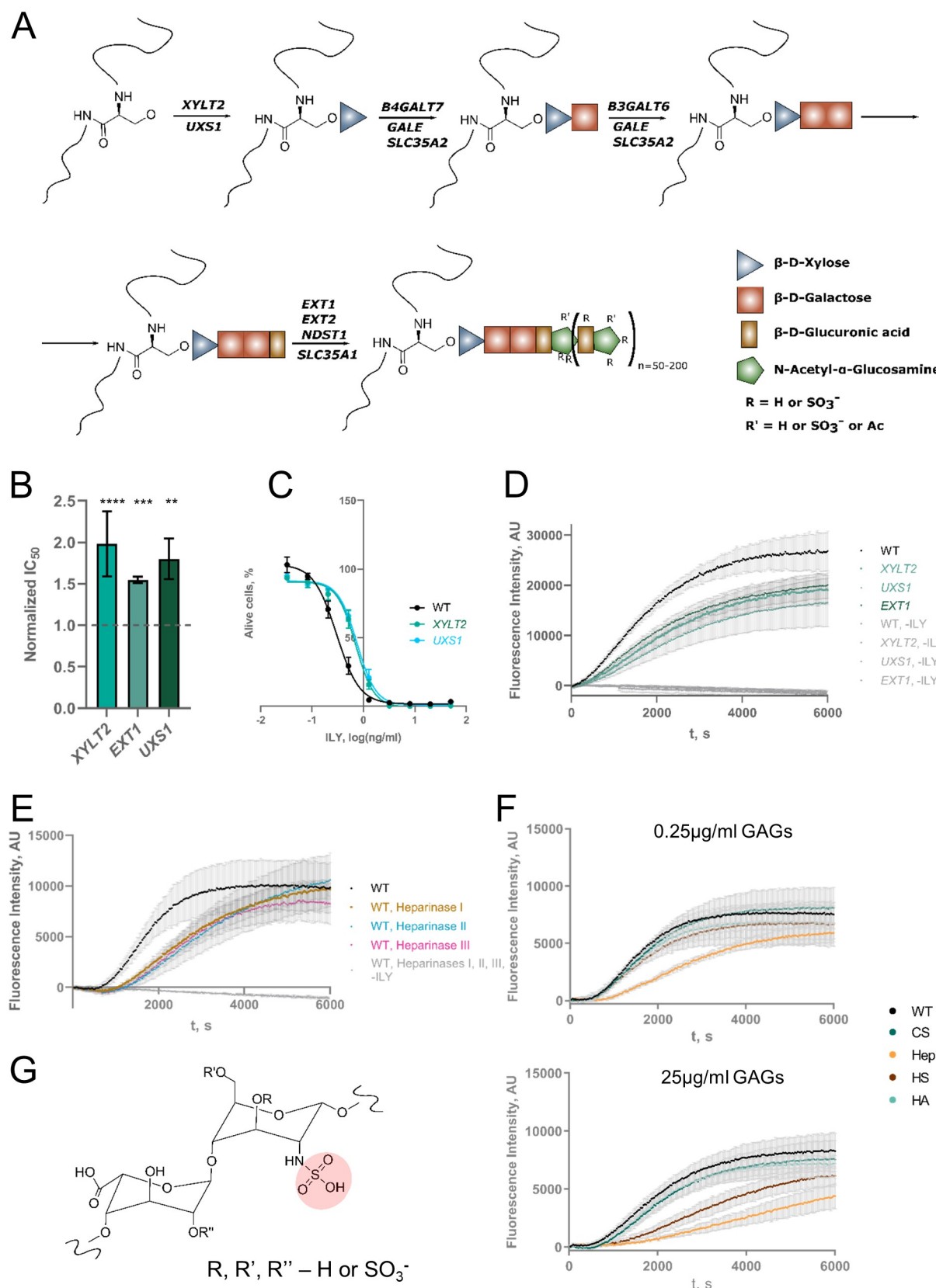

**Fig 4. Heparan sulfate knock-out inhibits the action of ILY. A**. Schematic representation of heparan sulfate synthesis. Genes listed on the arrows were identified in this study. **B**. Resistance conferred by the knock-out of select heparan sulfate pathway genes. Values were normalized by dividing the raw IC$_{50}$ value by the IC$_{50}$ value of ILY in WT HAP-1 cells. The dashed line shows the WT IC$_{50}$ value. n = 3, error bars represent standard deviations. p-values were calculated using two-tailed t-test; $*-<0.05$; $**-<0.01$, $***-<0.001$; $****-<0.0001$. **C**. Lysis profiles of cell lines with XYLT2 and UXS1 KOs. n = 3, error bars represent standard deviations. **D**. Qualitative kinetics of the ILY pore formation on the cells bearing heparan sulfate pathway KOs. Higher fluorescence intensity corresponds to more propidium iodide entering pores formed by ILY and is a surrogate measurement of the speed at which pores form and cells are lysed. n = 3, error bars represent standard deviations. **E**. Qualitative kinetics of the ILY pore formation on the cells treated with heparinase enzymes. Higher fluorescence intensity corresponds to more propidium iodide entering pores formed by ILY and is a surrogate measurement of the speed at which pores form and cells are lysed. n = 3, error bars represent standard deviation. **F**. Qualitative kinetics of the ILY pore formation on the WT cells in the presence of different glycosaminoglycans (GAGs). WT–no GAG added; CS–chondroitin sulfate; Hep–heparin; HS–heparan sulfate; HA–hyaluronic acid. Higher fluorescence intensity corresponds to more propidium iodide entering pores formed by ILY and is a surrogate measurement of the speed at which pores form and cells are lysed. Heparin (and heparan sulfate, at higher concentrations) competitively inhibits ILY. n = 3, error bars represent standard deviations. **G**. Structure of heparin or heparan sulfate repeating disaccharide unit. Out of all GAGs tested, heparin and heparan sulfate are the only GAGs that can be N-sulfonylated at the amine group of the glucuronic acid (although N-sulfate can be substituted by acetyl- moiety). The unique sulfate group is highlighted in pink.

ILY concentrations are much lower, and therefore, strategies such as using HS to increase the rate of cell lysis could have evolved. Hence, the host may use heparin as a decoy to gather up ILY or viruses to prevent them from attaching to the cell surface and, ultimately, to temper cell lysis or infection. Indeed, HS has often been identified as a cellular receptor or co-receptor for various toxins and viruses[38, 39].

## Glucosylceramides are key players in the membrane organization required for intermedilysin-mediated cell lysis

Because knock-out of genes form the UDP-sugar metabolism, GALE, and UGP2, elicits a significantly higher resistance than knock-out of the genes from the heparan sulfate pathway (Fig 2A), only part of the resistance due to UDP-sugar depletion can be ascribed to HS depletion, and additional mechanisms must be involved. Indeed, one of the major hits from our screen is *UGCG*.

UGCG synthesizes glucosylceramide (GluCer), transferring UDP-glucose to ceramide. This core serves as a basis for the synthesis of all ganglio- series of gangliosides (S7A Fig) in addition to other glycolipids sharing lactosylceramide as a core, such as Gb3, which is the receptor for Shiga toxin[27, 40]. Thus we reasoned that cells with *UGCG* or related pathway knock-out would fail to produce GM1 ganglioside in the membrane. Cholera toxin β subunit (CTB) binds to GM1, and labeled CTB can thus be used to stain GM1 in membranes[41]. We stained GM1 in the membrane of cell lines containing knock-out of validated hit genes involved in the ganglioside synthesis pathway (*UGCG*, *SLC35A2*). These knock-out lines lost their ability to bind CTB. UGCG is directly responsible for the synthesis of gangliosides, GM1 included, and SLC35A2 is a carrier protein that transports UDP-galactose to the ER, where GM synthesis takes place. Although B4GALT5 is also involved in ganglioside synthesis, it was not validated from our screen. Additionally, *B4GALT5* knock-out cells still contained GM1 in their membrane (S7C Fig). B4GALT5 adds UDP-galactose to glucosylceramide[42], but this function may be taken over by B4GALT6, a counterpart of B4GALT5, which explains our results. We also observed a lack of GM1 in the membrane of cell lines containing knock-out of genes involved in UDP-sugar metabolism, such as *UGP2* and *GALE* (S7B Fig), which provide the substrates for the genes *UGCG* and *B4GALT5* in this pathway. Because in our screen, we only identified *UGCG*, which is required to synthesize glucosylceramide (GluCer), we wanted to confirm the GM1 staining results by quantifying GluCer in *UGP2* and *GALE* knock-out cell lines directly. We also measured lactosylceramide (LacCer) levels to see whether it was also depleted due to the loss of UDP-glucose and UDP-galactose. Mass spectrometry revealed that

both *UGP2* and *GALE* knock-out cell lines contained a reduced amount of LacCer. In contrast, *GALE* knock-out cells were enriched, and *UGP2* knock-out cells were depleted for hexosylceramides (HexCer) (S8 Fig). Although it is impossible to differentiate between galactosyl- and glucosylceramides from mass spectrometry, we believe that the observed enrichment in hexosylceramides in the *GALE* knock-out cells is the result of increased glucosylceramides (which, without UDP-galactose, are useless for the production of GM1, which we visualized using CTB), because *GALE* knock-out cells cannot produce UDP-galactose. In contrast, the *UGP2* knock-out cell line can produce neither galactosyl-nor glucosylceramides and hence is found depleted in hexosylceramides (S8B Fig).

Another interesting hit from our ILY screen that is possibly involved in the ceramide metabolism is *TM9SF2*. This gene was only very recently discovered to be involved in the sphingolipid[24, 27] and possibly ceramide metabolism[43]. In agreement with this function, we observed that knock-out cells failed to bind CTB (S7C Fig). Furthermore, TM9SF2 is also involved in heparan sulfate synthesis, underpinning its role in several glycosylation pathways and membrane composition[44, 45].

In general, these results demonstrate that knock-out of genes resulting in the changes in the membrane's lipid constitution confers resistance to ILY. Specifically, loss of glucosyl and lactosylceramides protects cells from ILY. However, because we did not identify hits in the ganglioside synthesis pathway downstream of LacCer synthesis, we believe that GluCer and LacCer *per se* are important for ILY, as their absence may change membrane organization and cholesterol accessibility and thus interfere with the toxin's ability to interact with the membrane. Indeed, it has been demonstrated that glucosylceramides can profoundly influence the biophysical properties of the membrane and help to segregate transient membrane domains[46].

## Pneumolysin depends on the asymmetric distribution of the plasma membrane phospholipids

To test whether the genes identified in the ILY screen are also involved in the mechanism of other CDCs, we cross-validated the hits with two other CDCs, namely vaginolysin from *Gardnerella vaginalis* (VLY) and pneumolysin from *Streptococcus pneumoniae* (PLY). These toxins share a common pore formation mechanism; however, they differ in cellular factors required to initiate it. VLY can form pores independently from CD59; however, in this case, CD59 acts as an enhancer, greatly increasing VLY-mediated cell lysis[6]. PLY, on the other hand, is entirely independent from CD59, and only cholesterol is necessary for pores to form.

As expected, the loss of either *CD59* or *PIGA* genes strongly inhibited VLY activity (Fig 5A), which, as mentioned above, is partially dependent on CD59[4, 6]. However, almost no other hits discovered in the ILY screen had any effect on VLY, except for *SREBF2* and *PDCD10* (Fig 5A).

PLY, on the other hand, showed more overlapping hits with ILY. Its activity was not affected by the loss of *CD59* (Fig 5B) but appeared more sensitive to the loss of cholesterol sequestration genes than VLY (Fig 5A). Furthermore, neither PLY nor VLY depended on heparan sulfates, making it a possibly unique feature for ILY (Fig 5).

Surprisingly, PLY was extremely sensitive to the loss of the *TMEM30A* gene as well as to knock-out of *C1orf43*, *C12orf49*, and *PDCD10* (Fig 5B). Recently, *C1orf43* has been implicated in regulating endocytosis[47], *C12orf49* has been shown to regulate cholesterol metabolism through the SREBF pathway[48], and *PDCD10* modulates apoptotic pathways and is important for the normal structure and function of Golgi[49]. The exact mechanism of how they are involved in PLY cytolysis remains to be elucidated, but the function of these hits seems logical.

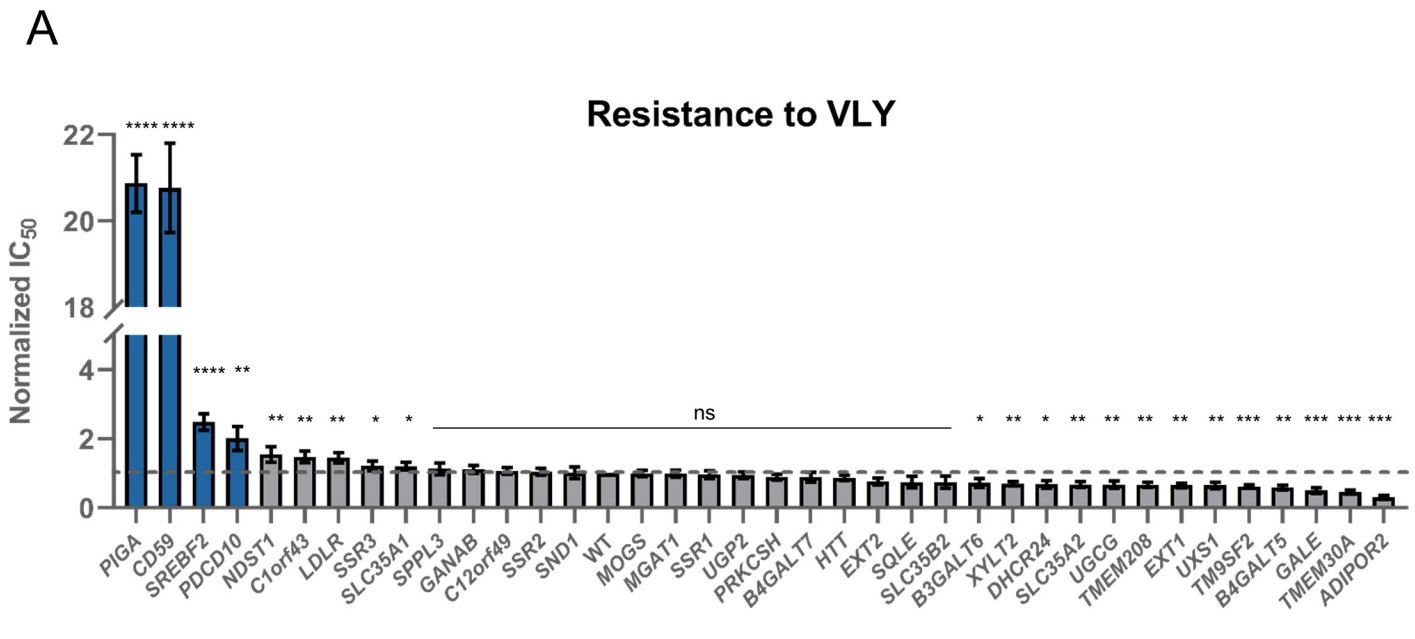

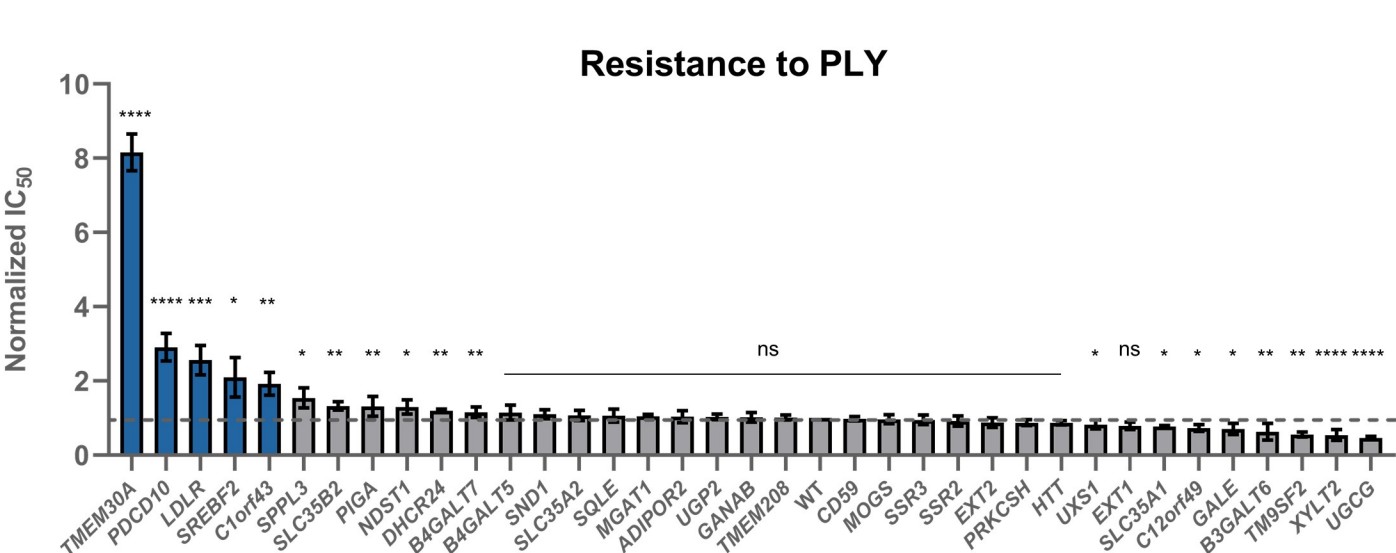

**Fig 5. Screening of ILY reveals information about other CDCs–vaginolysin and pneumolysin. A**. Resistance to vaginolysin (VLY) conferred by the knock-out of all tested genes. Values were normalized by dividing the raw $IC_{50}$ value by the $IC_{50}$ value of ILY against WT HAP-1 cells. The dashed line shows the WT $IC_{50}$ value. n = 3, error bars represent standard deviations. p-values were calculated using two-tailed t-test; *–<0.05; **–<0.01, ***–<0.001; ****–<0.0001. Significance was calculated in relation to the WT cell line. Top hits are highlighted in blue. **B**. Resistance to pneumolysin (PLY) conferred by the knock-out of all tested genes. Values were normalized by dividing the raw $IC_{50}$ value by the $IC_{50}$ value of ILY against WT HAP-1 cells. The dashed line shows the WT $IC_{50}$ value. n = 3, error bars represent standard deviations. p-values were calculated using two-tailed t-test; *–<0.05; **–<0.01, ***–<0.001; ****–<0.0001. Significance was calculated in relation to the WT cell line. Top hits are highlighted in blue.

*TMEM30A* (CDC50A) encodes the beta subunit of phospholipid flippase (P4-ATPase) and is likely of great importance in membrane organization and composition. These flippases maintain asymmetric distribution of the phospholipids by regulating the translocation of phosphatidylserine from the outer to the inner leaflet of the plasma membrane[50–52]. Loss-

of-function mutation in *TMEM30A* has recently been found to lead to an accumulation of lipid chemotherapy drugs and better treatment outcomes[53].

The reason why the loss of *TMEM30A* gene impacts PLY more than ILY and VLY remains to be elucidated. Still, we hypothesize that the electrostatic surfaces of different toxins may play a role (S9 Fig). PLY is mostly negatively charged, especially in the membrane-binding domain, which suggests that the accumulation of the negatively charged phosphatidylserines in the membrane makes it harder for the toxin to close in on the membrane cholesterol. ILY and VLY are much more positively charged, but in the case of these toxins, their dependence on CD59 may be a more important factor and hence the overall charge of the membrane may be less important.

In conclusion, we demonstrate that, in addition to the well-known CD59 receptor, ILY cytotoxicity also depends on different cellular pathways that have previously gone unnoticed. Most notably, we found that heparan sulfates facilitate the activity of ILY, and heparin can be used to interfere with it. Additionally, glucosyl- and/or lactosylceramides were critical, most likely by influencing the membrane structure. Lastly, we found that *TMEM30A*, identified in the ILY screen, is a major determinant in PLY cytotoxicity, suggesting that membrane charge plays an important role for PLY. Moreover, our findings illustrate that cholesterol-dependent cytolysins can be applied as tools for studying membrane dependencies.

## Materials and methods

### Cell culture

HEK293T cells were cultivated in DMEM medium (Gibco) supplemented with 10% heat-inactivated fetal bovine serum (Biowest) and 20 µg/ml gentamicin. HAP1 cells were cultivated in IMDM medium supplemented with 10% heat-inactivated fetal bovine serum (Biowest) and 20 µg/ml gentamicin. Cells were grown in humidified incubators at 37˚C and 5% $CO_2$. For complementation experiments, IMDM (Gibco) was further supplemented with 5 g/L D-galactose dissolved in PBS and filtered using a 0.22 µm filter. Cells are periodically tested for mycoplasma infections in the lab, and the cells used for this study were mycoplasma-free.

### Toxins

Recombinant intermedilysin, pneumolysin, and vaginolysin were a kind gift from Prof. Aurelija Zvirbliene and Dr. Milda Zilnyte. Briefly, the recombinant toxins were prepared as follows. The genes encoding respective CDC toxins lacking putative signal sequences were cloned into pET-28(+) vector fusing them to N-terminal 6xHis-tag, followed by thrombin protease site. *E. coli* BL21(DE3) strain was transformed with the constructs and protein expression induced by the addition of IPTG. Cells were then collected by centrifugation, disrupted by sonication, and centrifuged again to separate cell debris. Toxins were purified from the soluble part of the lysate using HisTrap FF and HisTrap SP columns (GE Healthcare), followed by size exclusion chromatography using Superdex 75 16/60 column (GE Healthcare). The 6xHis-tag was removed by incubating the purified protein with thrombin protease. Proteins were then stored at -20˚C.

### Production of TKOv3 library lentiviral vectors

TKOv3 lentiviral vector production was performed as described previously[28]. Briefly, HEK293T cells were seeded at the $9 \times 10^6$ cells per T150 cell culture flask in 20 ml of the medium. The next day, cells were co-transfected with lentiviral packaging vectors psPAX2 (4,8 µg), pMDG.2 (3,2 µg), and TKOv3 lentiCRISPR plasmid library (8 µg), using X-

tremeGene 9 transfection reagent (Roche) according to manufacturer's instructions. Toronto human knock-out pooled library (TKOv3) was a gift from Jason Moffat (Addgene #90294). After 24h, the medium was exchanged to serum-free DMEM, supplemented with 1% BSA. Medium containing a virus was collected after another 24h. The medium was spun down in a centrifuge at 1000g for 5 minutes. Lentivirus containing media was aliquoted and frozen at -80˚C. Functional viral titers on HAP1 cells were determined by transducing cells with different volumes of the virus-containing medium in the presence of 8 μg/ml polybrene. After 24h, the medium was changed to a medium supplemented with 1 μg/ml puromycin. After selecting cells for 48h, they were counted, and multiplicity of infection was calculated by comparing survival percentages of transduced and non-transduced cells.

## Genome-wide knock out screening in HAP1 cells

The screening was performed as described previously (Moffat, 2017). Briefly, $75 \times 10^6$ cells were transduced with the TKOv3 lentiviral library at MOI $\approx 0.3$ to maintain coverage of >200-fold. After 24h, we began cell selection by changing the medium to the one containing 1 μg/ml puromycin. After 48h, cells were pooled and split in two or three replicates of $20 \times 10^6$ cells each. At this point, we started selection with the toxins. We used 2 ng/ml ILY (we increased the concentration to 10 ng/ml when cells became resistant) or 2 ng/ml Diptheria toxin (Sigma Aldrich) per T150 flask containing 20 ml medium and $5 \times 10^6$ cells. Cells were passaged every 3 days or as needed. When passaging, cells from the flasks of the same replicate were pooled together, and $20 \times 10^6$ cells seeded once again with the toxin of interest. $20 \times 10^6$ cells were collected from each replicate every other splitting or, in case of ILY, before increasing toxin concentration and at the conclusion of the screen. Furthermore, $20 \times 10^6$ cells were collected before the start of the selection. We extracted genomic DNA using the QIAamp Blood Maxi Kit (Qiagen). sgRNA harboring integrated lentiviral sequences were amplified using primers containing Illumina i5 and i7 barcodes. Sequencing libraries were sequenced commercially on an Illumina HiSeq3000 machine.

Sequencing results were analyzed using the CRISPRCloud server[54] to obtain both gene and guide level statistics. We compared sgRNA distributions between non-selected samples and toxin-selected samples. Genes for which at least three out of four guides were enriched ≥2.5-fold with a p-value of <0.01 were considered hits. All screens were performed twice.

## Knock-out cell line generation

For the hit validations, we used guides from Brunello library[30] (S2 Table) against the gene of interest. Guides were ordered as single-stranded DNA oligonucleotides (IDT) with 3' and 5' appended sequences to facilitate ligase independent cloning. Guide oligonucleotides were hybridized with the vector and a universal primer by heating to 95˚C and slowly cooling to room temperature. Competent NEB5α cells were transformed with the constructs and spread on LB agar Petri dishes containing 100 μg/ml ampicillin. Single colonies were grown out in liquid LB medium, supplemented with 100 μg/ml ampicillin. Plasmid DNA was extracted using the NucleoSpin Plasmid Kit (Machery-Nagel). Guide cloning success was validated by Sanger sequencing (Macrogen).

To knock out particular genes, HAP1 cells were seeded in 6-well plates with $2 \times 10^5$ cells per well. Vectors containing four different guides against the same gene were pooled in equimolar ratios. Cells were transfected with guide pools using Turbofectin 8.0 transfection reagent (OriGene), according to manufacturer's instructions. 24h post-transfection, media was changed to the one containing 1 μg/ml puromycin. After 48h selection, cells were split, and $2 \times 10^5$ cells were seeded in a single well of a 6-well plate for outgrowth. 48h later, cells were seeded at 2 x

$10^5$ cells per well in 6-wells of a 6-well plate and selected with 2-fold dilutions of ILY. Cells surviving the highest toxin concentration were expanded and were considered as a knock-out cell line in subsequent experiments. The same procedure was performed with WT HAP1 cells by transfecting empty vectors to exclude non-gene-specific resistance to ILY. These toxin-selected WT HAP1 cells were used as a control in all subsequent experiments.

## Fluorescence-activated cell sorting (FACS)

WT HAP1 or single-gene knock-out cells were detached using 0.05% Trypsin-EDTA (Gibco) and resuspended in 10% FBS containing medium. Subsequently, they were washed with PBS containing 1% FBS and incubated for 30 min at 37˚C with a primary anti-CD59 antibody (OV9A2) conjugated with APC (Thermo Scientific). Cells were washed twice to remove unbound antibody. Labeled cells were analyzed on the FACS Celesta flow cytometer. Data were analyzed using FlowJo software. Live cells were gated according to areas of side and forward scatter. WT cell and *PIGA* or *CD59* cell median fluorescence intensities in the APC channel were compared. Analyses were performed from at least two independent technical replicates for each of the cell lines, each containing at least 15000 gated events.

## MTS cell proliferation assay

HAP1 WT or single-gene knock-out cells were seeded in 96-well plates at $2 \times 10^4$ cells per well per 200 μl in at least 3 replicates. ILY, VLY, or PLY (CDC) serial dilutions in culture medium were added to the cells the next day. Cells were incubated with CDCs for 1h at 37˚C. Subsequently, 20 μl of MTS reagent was added to each well. After incubating cells at 37˚C for 4-6h, plates were scanned using Tecan Saphire 2 plate reader. Absorption at 490 nm was measured. Cells without toxin were used as a negative control, medium without cells–a positive control. Data were plotted and fitted with a non-linear regression model in GraphPad Prism 8. $IC_{50}$ values were used to compare cell line resistances to ILY.

## GM1 staining and fluorescence microscopy

$1 \times 10^4$ HAP1 WT of single-gene knock-out cells were seeded in a well of an 8-well μ-slide (Ibidi). The next day, GM1 on the cell surface was visualized using Vybrant Alexa Fluor 594 Lipid Raft labeling kit (Thermo Scientific). Briefly, to each well, we added 2 μl of the 1 mg/ml of fluorescently labeled Cholera toxin subunit B (CTB), which binds GM1 molecules, and incubated cells for 15min at 37˚C. Afterward, cells were washed carefully with PBS several times. Next, cells were incubated with anti-CTB rabbit serum for 15mins in 200 μl of PBS at 37˚C. Subsequently, cells were washed several times with PBS and imaged using Leica SP5 confocal microscope with a CX PL APO 63× (NA 1.2) water-immersion objective.

## Lipid mass spectrometry

**Lipid extraction.**   700 μl of homogenized cells were mixed with 800 μl 1 N $HCl:CH_3OH$ 1:8 (v/v), 900 μl $CHCl_3$ and 200 μg/ml of the antioxidant 2,6-di-tert-butyl-4-methylphenol (BHT; Sigma Aldrich). 3 μl of SPLASH LIPIDOMIX Mass Spec Standard (#330707, Avanti Polar Lipids) was spiked into the extract mix. The organic fraction was evaporated using a Savant Speedvac spd111v (Thermo Fisher Scientific) at room temperature, and the remaining lipid pellet was stored at—20˚C under argon gas.

**Mass spectrometry.**   Just before mass spectrometry analysis, lipid pellets were reconstituted in 100% ethanol. Lipid species were analyzed by liquid chromatography-electrospray ionization tandem mass spectrometry (LC-ESI/MS/MS) on a Nexera X2 UHPLC system

(Shimadzu) coupled with hybrid triple quadrupole/linear ion trap mass spectrometer (6500 + QTRAP system; AB SCIEX). Chromatographic separation was performed on a XBridge amide column (150 mm × 4.6 mm, 3.5 μm; Waters) maintained at 35°C using mobile phase A [1 mM ammonium acetate in water-acetonitrile 5:95 (v/v)] and mobile phase B [1 mM ammonium acetate in water-acetonitrile 50:50 (v/v)] in the following gradient: (0–6 min: 0% B ➜ 6% B; 6–10 min: 6% B ➜ 25% B; 10–11 min: 25% B ➜ 98% B; 11–13 min: 98% B ➜ 100% B; 13–19 min: 100% B; 19–24 min: 0% B) at a flow rate of 0.7 ml/min which was increased to 1.5 mL/min from 13 minutes onwards. SM, CE, CER, DCER, HCER, LCER were measured in positive ion mode with a precursor scan of 184.1, 369.4, 264.4, 266.4, 264.4, and 264.4, respectively. TAG, DAG, and MAG were measured in positive ion mode with a neutral loss scan for one of the fatty acyl moieties. PC, LPC, PE, LPE, PG, LPG, PI, LPI, PS, and LPS were measured in negative ion mode by fatty acyl fragment ions. Lipid quantification was performed by scheduled multiple reactions monitoring (MRM), the transitions being based on the neutral losses, or the typical product ions as described above. The instrument parameters were as follows: Curtain Gas = 35 psi; Collision Gas = 8 a.u. (medium); IonSpray Voltage = 5500 V and −4,500 V; Temperature = 550°C; Ion Source Gas 1 = 50 psi; Ion Source Gas 2 = 60 psi; Declustering Potential = 60 V and −80 V; Entrance Potential = 10 V and −10 V; Collision Cell Exit Potential = 15 V and −15 V.

## Data analysis

Peak integration was performed with the MultiQuant software version 3.0.3. Lipid species signals were corrected for isotopic contributions (calculated with Python Molmass 2019.1.1) and were normalized to internal standard signals. Unpaired T-test p-values and FDR corrected p-values (using the Benjamini/Hochberg procedure) were calculated in Python StatsModels version 0.10.1.

## Propidium iodide pore formation assay

HAP1 WT or heparin sulfate pathway knock-out cell lines were seeded in black 96-well plates at $10^4$ cells per well. In the morning, the following mixture was prepared: 0.2 ng/ml ILY, 1 μg/ml propidium iodide (Thermo Scientific) in PBS. For some experimental conditions, heparin, heparan sulfate, chondroitin sulfate, or hyaluronic acid (Sigma-Aldrich) solution at the desired concentrations was also added to the mixture. The cell culture medium was then carefully removed from the cells and replaced with 200 μl of the ILY and PI mixture. Plates were analyzed on a Tecan Saphire 2 instrument, reading the plate every 30s, with 490 nm excitation and 630 nm emission wavelengths. All values starting with the second were subtracted from the first one and plotted as time series.

For some experiments, cells were treated with Heparinase I, II, or III enzymes from *Bacteroides eggerthii* (NEB). HAP1 WT cells were seeded in black 96-well plates at $10^4$ cells per well. The next day, the cell culture medium was then carefully removed, and 100 μl of 1x heparinase buffer added to cells, followed by the addition of 10U of the desired heparinase enzyme. Cells were then incubated for 1h at 37°C. Next, 100 μl of the 2x ILY and propidium iodide mixture was added. Plates were then analyzed as described above.

## Supporting information

**S1 Table. Guide RNA enrichments from all of the screens in the study.** The table displays the amount of reads of each guide in the screen.
(XLSX)

**S2 Table. List of guide RNAs used for hit validation.** The table shows sgRNA sequences used for gene knock-out during hit validation. Additionally, it includes the overhangs used for cloning into vectors and final oligonucleotides.
(XLSX)

**S1 Fig. Diphtheria toxin genome-wide CRISPR screen. A**. Volcano plot of a representative DT genome-wide CRISPR screen replicate. Y-axis displays the significance of the hit, while the X-axis represents $Log_2$-fold change in the average number of guide reads (of all four guides) when compared with non-selected cells. **B**. Results of two independent DT CRISPR knock-out screens. Axes represent $Log_2$-fold changes in the average number of guide reads (of all four guides) when compared with non-selected cells from two independent replications of the screen. We identified all of the genes involved in the diphthamide biosynthesis and the toxin receptor–HBEGF.
(TIF)

**S2 Fig. 10 ng/ml intermedilysin genome-wide CRISPR knock-out screen. A**. Volcano plot of the representative ILY genome-wide CRISPR screen after the second round of selection with 10 ng/ml ILY. Axes represent $Log_2$-fold changes in the number of the average number of guide reads (of all four guides) when compared with non-selected cells from two independent replications of the screen. We identified most of the genes in GPI anchor synthesis and attachment cascades as well as the receptor of ILY–CD59. **B**. Results of two independent CRISPR knock-out screens after the second round of selection with 10 ng/ml ILY. Axes represent $Log_2$-fold changes in the average number of guide reads (of all four guides) when compared with non-selected cells from two independent replications of the screen.
(TIF)

**S3 Fig. Guide RNA enrichment of every hit in the ILY screens.** Different colors of dashes represent $Log_2$-fold enrichments of guides from different independent ILY screens—guide RNA enrichment from the first screen depicted in blue, guide RNA enrichment from the second screen depicted in red.
(TIF)

**S4 Fig. UDP-sugar synthesis from glucose pathway.** Genes listed on the arrows were identified in this study.
(TIF)

**S5 Fig. UDP-sugar synthesis from glucose pathway.** Qualitative kinetics of the ILY pore-formation on the WT cells with added different glycosaminoglycans (GAGs) at various concentrations. WT–no GAG added; CS–chondroitin sulfate; Hep–heparin; HS–heparan sulfate; HA–hyaluronic acid. Higher fluorescence intensity corresponds to more propidium iodide entering pores formed by ILY and is a surrogate measurement of the speed at which pores form and cells are lysed. Heparin and heparan sulfate, at higher concentrations, competitively inhibit ILY. However, at highest concentrations tested other GAGs inhibit ILY as well, suggesting non-specific electrostatic interactions. Bottom left panel represents a control experiment, with no ILY added. n = 3, error bars represent standard deviations.
(TIF)

**S6 Fig. Sulfate ions (red and yellow) present in the first crystal structure of ILY.** PDB ID: 1S3R.
(TIF)

**S7 Fig. Some KO cell lines lose the ability to bind CTB. A**. Schematic representation of GM1 ganglioside biosynthesis synthesis. Genes listed on the arrows were identified in this study. **B**.

Staining of the cells bearing UDP-sugar gene knock-out with cholera toxin B (CTB) labeled with AlexaFluor594. CTB binds GM1 gangliosides present in the plasma membrane. UGP2 and GALE cells did not stain with the CTB. **C**. Staining of the cells bearing listed gene knock-out with cholera toxin B (CTB) labeled with AlexaFluor594. CTB binds GM1 gangliosides present in the plasma membrane. *UGCG*, *SLC35A2*, and *TM9SF2* knock-out cells did not stain with CTB.
(TIF)

**S8 Fig. Changes in the amount of hexosyl- and lactosylceramides present in the knock-out cell lines. A**. Changes in the amount of hexosylceramide present in *GALE* and *UGP2* knock-out cell lines. n = 3, error bars represent standard deviation. p-values were calculated using two-tailed t-test; *–<0.05; **–<0.01, ***–<0.001; ****–<0.0001. **B**. Changes in the amount of lactosylceramide present in *GALE* and *UGP2* knock-out cell lines. n = 3, error bars represent standard deviation. p-values were calculated using two-tailed t-test; *–<0.05; **–<0.01, ***–< 0.001; ****–<0.0001.
(TIF)

**S9 Fig. Surface charge distribution of CDCs.** Representation of surface charge distribution of PLY, ILY, and VLY in an overall protein and the bottom, membrane interacting surface. In contrast to VLY and ILY, PLY is more negative overall as well as in the membrane-interacting domain. Representations were generated from the following structures: PDB IDs: 1SR3; 5CR6; 5IMY. Charge distributions were calculated using the Poisson-Boltzmann solver plugin in Pymol.
(TIF)

## Acknowledgments

We kindly thank Milda Zilnyte and Aurelija Zvirbliene for providing ILY, VLY, and PLY proteins. Jonas Dehairs and Johan Swinnen for performing lipid mass spectrometry. Furthermore, GD wishes to thank Milda Zilnyte, Migle Kazlauskiene, Irmantas Mogila, and Els Vanstreels for helpful discussions and comments on the manuscript.

## Author Contributions

**Conceptualization:** Gediminas Drabavicius, Dirk Daelemans.

**Data curation:** Gediminas Drabavicius.

**Formal analysis:** Gediminas Drabavicius, Dirk Daelemans.

**Funding acquisition:** Dirk Daelemans.

**Investigation:** Gediminas Drabavicius.

**Methodology:** Gediminas Drabavicius.

**Supervision:** Dirk Daelemans.

**Validation:** Gediminas Drabavicius.

**Visualization:** Gediminas Drabavicius.

**Writing – original draft:** Gediminas Drabavicius.

**Writing – review & editing:** Gediminas Drabavicius, Dirk Daelemans.

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
