## [Decision Letter · Decision Letter 0]

15 Sep 2020

Dear Dr Daelemans,

Thank you very much for submitting your Research Article entitled 'Intermedilysin cytolytic activity depends on heparan sulfates and membrane composition' to PLOS Genetics. Your manuscript was fully evaluated at the editorial level and by independent peer reviewers. The reviewers appreciated the attention to an important problem, but raised some substantial concerns about the current manuscript. Based on the reviews, we will not be able to accept this version of the manuscript, but we would be willing to review again a much-revised version. We cannot, of course, promise publication at that time.

If you decide to revise the manuscript for further consideration at PLOS Genetics, please aim to resubmit within the next 60 days, unless it will take extra time to address the concerns of the reviewers, in which case we would appreciate an expected resubmission date by email to plosgenetics@plos.org.

[LINK]

We are sorry that we cannot be more positive about your manuscript at this stage. Please do not hesitate to contact us if you have any concerns or questions.

Yours sincerely,

Gregory S. Barsh

Editor-in-Chief

PLOS Genetics

Gregory Copenhaver

Editor-in-Chief

PLOS Genetics

Reviewer's Responses to Questions

**Comments to the Authors:**

Reviewer #1: Here the authors carried out genome-wide CRISPR knockout screens for a representative cholesterol-dependent cytolysins (CDC), intermedilysin (ILY). The screen successfully identified the known receptor CD59, and many weak hits. The authors validated many of them and determined that heparan sulfates facilitate the kinetics of forming ILY pore on cell surfaces, likely by mediating toxin attachment onto cells, and disrupting glucosylceramide pathway could also reduce ILY activity. Finally, they examined two other related CDCs and found that one (pneumolysin) showed reduced activity when TMEM30A, a phospholipid flippase is disrupted. These findings are novel and interesting, adding new insight to our understanding of CDCs. The writing and figure organization however need to be further improved.

Major issues:

The specificity of heparan sulfate for ILY can be further explored to determine whether it is the sulfate group that is responsible for ILY attachment, for instance, by examining competition with other sulfated GAGs and non-sulfated GAG in Fig. 3G.

Minor issues:

Line 84: One of the early successful CRISPR-Cas9 screen on bacterial toxin is PMID: 27680706 (Tao et al.).

Line 98: use ILY, no need to use intermedilysin after the first time.

Line 104: there should be a space between number and unit. Please check.

Line 113: Diphtheria toxin is more commonly called DT, not DFT.

Line 227 and Suppl. Fig. 7, UGCG is required not just for gangliosides, but also other glycolipids such as Gb3. It was also a hit in recent Shiga toxin screens.

Line 635: The title needs to be changed. E.g. Results of two independent ILY CRISPR KO screens.

HS pathway is frequently identified in screens for many pathogens, viruses, and also for bacterial toxins (e.g. PMID: 31160825). The authors can add a short discussion on it.

Fig. 1D: is the unit ng/ml? there is no need to go up to 150 on y-axis.

Fig. 2A, 3A: It would be better to remove these panels, and directly note the hits on an enlarged Fig. 1C.

Fig. 3E: Is the XYLT2 the same data as in panel B? No need to replot it here.

Fig. 4C: this panel can be moved to supplementary figures, as evidence for membrane charge contribution is weak.

Move supplementary Fig. 6 and 13 into main figures – these data represent lots of work and are interesting.

Reviewer #2: The current study describes a genome-wide Crispr/Cas9 screen of cellular factors responsible for intermedilysin membrane binding/cell lysis.

I quite liked this paper – the study is well-done, interesting, logical and well-written. The authors have identified a number of novel genes responsible for ILY lysis, and the targets have been carefully validated.

As with any other screen, the current paper raises many potentially interesting questions for future studies.

There two points that I feel the authors should address.

1. Since CD59 is a plasma membrane glycoprotein, it should be demonstrated that none of the positive hits alter its surface levels – altered glycosylation may influence the stability of CD59 and lipid redistribution may influence its surface exposure. What is the glycosylation status of CD59, ie does it contain heparan sulfate moiety? If so, does it regulate ILY-CD59 interaction? This can be potentially validated by mutating the O-linked glycan sites and re-expressing the mutant protein in CD59-knockout cells.

2. How do the authors distinguish ILY membrane binding and failure to lyse the cells, from the lack of membrane binding? This can be assessed by the Western immunoblotting of membrane-bound ILY, if there are antibodies available or the protein is tagged for purification (it is unclear how ILY was purified), or by flow cytometry. The reason for this question is a recent study by Rudd-Schmidt et al (2019) where another pore-forming protein, perforin, was shown to bind to PS but could not lyse the cells.

**Have all data underlying the figures and results presented in the manuscript been provided?**

Reviewer #1: Yes

Reviewer #2: Yes

PLOS authors have the option to publish the peer review history of their article (what does this mean?). If published, this will include your full peer review and any attached files.

Reviewer #1: No

Reviewer #2: No

---

## [Decision Letter · Decision Letter 1]

22 Dec 2020

Dear Dr Daelemans,

Thank you very much for submitting your Research Article entitled 'Intermedilysin cytolytic activity depends on heparan sulfates and membrane composition' to PLOS Genetics.

The revised manuscript was seen by the original 2 reviewers. As you will see, both reviewers are generally positive with minor concerns; additionally, reviewer #2 identifies an important issues that we ask you address in a hopefully final revision that will be evaluated by the reviewer.

We therefore ask you to modify the manuscript according to the review recommendations. Your revisions should address the specific points made by each reviewer.

We hope to receive your revised manuscript within the next 60 days. If you anticipate any delay in its return, we would ask you to let us know the expected resubmission date by email to plosgenetics@plos.org.

[LINK]

Yours sincerely,

Gregory S. Barsh

Editor-in-Chief

PLOS Genetics

Gregory Copenhaver

Editor-in-Chief

PLOS Genetics

Reviewer's Responses to Questions

**Comments to the Authors:**

Reviewer #1: The authors have done a great job revising the manuscript and I only have a few minor edits:

Line 28: Cas9.

Line 36: delete “additionally”.

Line 41: delete “however”.

Line 89: delete “however”.

Line 252: delete “recently identified as” and change it to “which is the receptor for Shiga toxin”.

Line 761: change the title for this figure legend to directly mention VLY and PLY.

Reviewer #2: I appreciate the authors' effort in answering my queries.

I have noticed though that there was no clear correlation between the effect of gene knockout on CD59 expression and cytolysis. The authors rightly pointed out that they could not distinguish the effect of altered N-glycosylation on CD59 surface expression from other (eg global) effects of gene knockouts.

I apologise for being a pain, but they can in fact resolve that conundrum, or at least make a step in that direction, by conducting a simple experiment. They should knock-out CD59 using crispr, and then re-express it (eg by viral transduction). What they'll have as a result is cells that produce a wide range of CD59 protein. Then they can then sort various populations of cells that would match their other cell lines in terms of CD59 expression levels. In addition, the authors can treat the cells with ILY and assess CD59 expression levels before and after adding ILY - that way they should see an enrichment of CD59 low/negative population etc.

The outcome of these experiments is that they will be able to determine the minimum level of CD59 that will affect ILY lytic activity and, potentially, uncouple the effect of their glycosylation genes on CD59 from other effects that they might cause. It is possible, for example, that the diminished activity of ILY with respect to UGP2-knockout cells has nothing to do with the extent of the reduction of CD59 expression, or they might find an excellent correlation.

**Have all data underlying the figures and results presented in the manuscript been provided?**

Reviewer #1: Yes

Reviewer #2: Yes

PLOS authors have the option to publish the peer review history of their article (what does this mean?). If published, this will include your full peer review and any attached files.

Reviewer #1: No

Reviewer #2: No

---

## [Decision Letter · Decision Letter 2]

27 Jan 2021

Dear Dr Daelemans,

We are pleased to inform you that your manuscript entitled "Intermedilysin cytolytic activity depends on heparan sulfates and membrane composition" has been editorially accepted for publication in PLOS Genetics. Congratulations!

Yours sincerely,

Gregory S. Barsh

Editor-in-Chief

PLOS Genetics

Gregory Copenhaver

Editor-in-Chief

PLOS Genetics

Comments from the reviewers (if applicable):

Reviewer's Responses to Questions

**Comments to the Authors:**

Reviewer #2: While I do not completely agree with the authors' response, I still like the paper and have no further queries.

**Have all data underlying the figures and results presented in the manuscript been provided?**

Reviewer #2: None

PLOS authors have the option to publish the peer review history of their article (what does this mean?). If published, this will include your full peer review and any attached files.

Reviewer #2: No

**Data Deposition**

http://datadryad.org/submit?journalID=pgenetics&manu=PGENETICS-D-20-01190R2

**Press Queries**

---

## [Editor Report · Acceptance letter]

8 Feb 2021

PGENETICS-D-20-01190R2 

Intermedilysin cytolytic activity depends on heparan sulfates and membrane composition 

Dear Dr Daelemans, 

We are pleased to inform you that your manuscript entitled "Intermedilysin cytolytic activity depends on heparan sulfates and membrane composition" has been formally accepted for publication in PLOS Genetics! Your manuscript is now with our production department and you will be notified of the publication date in due course.

With kind regards,

Alice Ellingham

PLOS Genetics

On behalf of:
